# Follicle-Stimulating Hormone Promotes the Development of Endometrial Cancer *In Vitro* and *In Vivo*

**DOI:** 10.3390/ijerph192215344

**Published:** 2022-11-20

**Authors:** Shuman Sheng, Wei Liu, Yafei Xue, Zhengwu Pan, Lanlan Zhao, Fei Wang, Xiaoyi Qi

**Affiliations:** 1Department of Obstetrics and Gynecology, Shandong Provincial Hospital Affiliated to Shandong First Medical University, 324 Jingwu Road, Jinan 250021, China; 2Department of Obstetrics and Gynecology, Shandong Provincial Hospital, Shandong University, 324 Jingwu Road, Jinan 250021, China

**Keywords:** FSH, endometrial cancer, organic pollutants, endocrine disruptors, AMPK

## Abstract

Endocrine disruptors as risk factors for endometrial cancer (EC) are positively correlated with serum follicle-stimulating hormone (FSH) levels. Additionally, increased FSH is associated with EC. However, its exact mechanism is not yet clear. Therefore, this study investigated how FSH affects the occurrence of EC. Using immunohistochemistry (IHC), immunofluorescence (IF), and Western blot (WB), we found that FSH receptor (FSHR) was expressed in both EC tissues and cell lines. To explore the effect of FSH on EC *in vitro*, Ishikawa (ISK) cells were cultured in different doses of FSH, and it was found that FSH could promote the proliferation and migration of ISK cells. Furthermore, the detection of key molecules of migration and apoptosis by WB showed that FSH promoted cell migration and inhibited apoptosis. Additionally, FSH decreased AMPK activation. To clarify the effect of FSH on EC *in vivo*, we subcutaneously planted ISK cells into ovariectomized mice and then gave two of the groups oestradiol (E2). In comparison with the OE (ovariectomy plus E2) and sham groups, the growth rates and weights of the tumors in the OE plus FSH group were significantly higher. The findings above suggest that FSH promotes the proliferation and metastasis of EC, providing a new strategy for the treatment of EC.

## 1. Introduction

Organic pollutants in the environment have negative effects on human health. High concentrations of organic compounds are reported to cause a dysfunction of the human reproductive and endocrine system, further leading to changes in sex hormone activity [1]. Follicle-stimulating hormone (FSH) is a gonadotropin secreted by the anterior pituitary gland, which functions by binding its receptor. The FSH receptor (FSHR), which is a member of the Class A/Rhodopsin subfamily of the G protein-coupled receptors (GPCRs), mediates the regulatory function of FSH [2]. For several years, the role of FSH was thought to be confined to gonadal tissues. However, in recent times, researchers have paid increasing attention to its extra-gonadal roles, such as the regulation of lipid synthesis [2], metabolic syndrome [3], and, especially, the promotion of tumor development and progression.

A high BMI, early menarche, late menopause, family history of endometrial cancer, polycystic ovarian syndrome, radiation exposure, and exposure to estrogen or synthetic estrogen compounds such as EDCs have been identified as risk factors for EC [4,5]. EDCs are primarily derived from environmental pollutants and preservatives. They can disrupt endogenous hormone metabolism by mimicking their function [6]. Increased levels of organic pollutants in the abdominal adipose tissue of patients with endometrial stromal sarcomas implies that organic pollutants have a tendency to bioaccumulate in human tissues and may contribute to their negative effects on cells [7]. Some organic pollutions, such as di (2-Ethylhexyl) phthalate, can affect the release of pituitary hormones and increase the level of FSH, which is one of the types of pathogenesis of premature ovarian failure [8]. Harlow et al. proved that organic pollutants are positively correlated with FSH in middle-aged menopausal transitional women [9], which may be associated with the development of EC.

Endometrial cancer (EC) is the most common gynecology cancer in higher socioeconomic development countries and the second most common female malignant disease in China [10]. The prevalence of EC has globally increased in recent times [11]. It is well known that EC mostly occurs in postmenopausal women, and postmenopausal hormonal changes are characterized by a decrease in estrogen and a significant increase in FSH. EC can be divided into estrogen-dependent and non-estrogen-dependent cancers, with the former accounting for the majority, and its incidence is closely correlated to estrogen [10]. For the treatment of EC, hormone therapy is suitable for early stages and well-differentiated young patients with endometrial cancer who need to preserve reproductive function. Patients with advanced disease are treated with chemotherapy and less frequently with hormone therapy [12]. Organic pollutants have been found to be positively correlated with FSH in middle-aged menopausal transitional women [9]. Furthermore, there are some potential links between endocrine disruptor compounds (EDCs) and estrogen-dependent diseases such as EC, endometriosis, and breast cancer [4,6]. Recently, several studies have suggested that FSH is associated with EC. As examples, a clinical study [13] indicated that a high FSH level is a risk factor for EC, and a basic study [14] discovered that FSH can motivate endometrial cancer cells to migrate; however, the specific mechanism is not clear.

The dysregulation of energy homeostasis in cancer is considered as an essential factor driving disease changes. In eukaryotic cells, AMP-activated protein kinase (AMPK) is vital for cellular energy balance [15]. It is well known that programmed cell death is mediated by several protein factors that include the BCL-2 protein family and the caspase group of cysteine proteases. An increase in BAX level can promote cell apoptosis. Many preclinical studies have confirmed that BCL-2-regulated apoptosis is associated with the chemosensitizing effect of metformin [16]. Metformin is widely known as an AMPK inhibitor and has been used to treat metabolic diseases such as diabetes, obesity, and cancer. Particularly, AMPK is involved in the budding and development of EC [17] and suppresses cell proliferation and metastasis through the mammalian target of the rapamycin (mTOR) [18]. In women with polycystic ovary syndrome, metformin further inhibits FSH activity by reducing FSH mRNA, and thus FSH stimulates aromatase expression and activity in human ovarian granulosa cells [19]. Metformin also has potential anticancer effects and inhibits EC proliferation in both KLE and HEC-1A cell lines by activating AMPK [18]. However, the FSH/AMPK pathway has not been reported in EC. In view of this, we explored the effect of FSH on EC in this paper. We hypothesized that increased FSH in menopausal women can promote EC proliferation and invasion.

## 2. Methods

### 2.1. Human Tissue and Tissue Microarray

The tissue microarray (TMA) was purchased from the Shanghai Outdo Biotech Company (Shanghai, China). A total of 34 endometrial cancer tissues and their corresponding 11 adjacent tissues were used. All tissues were confirmed by histopathological examination. None of the EC patients coexisted with other malignancies or received adjuvant treatment including chemotherapy and/or radiotherapy. All procedures were approved by the Ethics Committee of Shandong Provincial Hospital. The patients’ clinical characteristics are described in Table 1.

### 2.2. Immunohistochemistry

Immunohistochemical (IHC) analysis was performed according to the manufacturer’s instructions. Briefly, the TMA was dewaxed in xylene and rehydrated with ethanol. Subsequently, sodium citrate (pH 6.0) was used to repair antigens and block non-specific binding by 3% H_2_O_2_. TMA was incubated at 4 °C overnight in a 1:50 concentration of rabbit-derived anti-FSHR antibody (ab150557, Abcam, Cambridge, UK). On the second day, secondary antibodies (ZSGB-Bio, Beijing, China) were used to incubate TMA, followed by 3,3′-diaminobenzidine (DAB) and hematoxylin staining. The image was obtained by a Pannoramic MIDI Digital slice scanner (3D HISTECH, Budapest, Hungary). The histochemistry scores (H-scores) were analyzed using Quant Center analysis software in the Pannoramic Viewer (3D HISTECH). We used the formula $histochemistry\;scores=\sum \left(PI\times I\right)$, where *PI* is the proportion of positive cells among all cells in the section and *I* corresponds to the color intensity of section. Dark brown, brownish yellow, light yellow, and blue nuclei are the different color intensity categories that were used to symbolize extremely positive, moderately positive, weakly positive, and negative, respectively.

### 2.3. Cell Lines and Cell Culture

The human endometrial cancer cell lines Ishikawa (ISK), KLE, and HEC-1A were bought from the Cell Bank of the Chinese Academy of Sciences (Beijing, China). HeLa was purchased from the Qilu Hospital Laboratory of Shandong University. ISK, KLE, and HeLa cells were incubated in DMEM high-glucose medium (HyClone, Logan, UT, USA) and supplemented with 10% fetal bovine serum (FBS, Biological Industries, Kibbutz Beit HaEmek, Israel) and 1% antibiotics (Macgene, Beijing, China). The HEC-1A cells were incubated in McCoy’s 5A medium (Gibco, Carlsbad, CA, USA) with 10% FBS and antibiotics. All cell lines were incubated in a humid environment of 37 °C, containing 5% CO_2_. Then, different concentrations of recombinant human follicle-stimulating hormone (FSH: 0 IU/L, 10 IU/L, 50 IU/L, and 100 IU/L [20], HOR-253, ProSpec, Rehovot, Israel) were added to the ISK cell culture medium.

### 2.4. Colony Formation Assay

The inoculation of 500 cells/well for each experimental group was carried out on 6-well culture plates. After being incubated for 10 days (37 °C, 5% CO_2_), the plates were then fixed with 4% paraformaldehyde for 20 min and stained with 0.1% crystal violet for 10 min at 25 °C. We used ImageJ software (National Institutes of Health, Bethesda, MD, USA) for the cell counts.

### 2.5. Cell Scratch Assay

The scratch wounds were made by a sterile pipette yellow tip when the ISK cells reached 90% confluence. Images of the wound closure in the same location were taken with a microscope (OLYMPUS) after 0, 24, and 48 h of incubation in a medium containing 1% fetal bovine serum. The cell migration rates were calculated using ImageJ software based on the healing area.

### 2.6. Transwell Migration Assay

A total of 5 × 10^4^ cells were added into the upper chamber of an 8 μm pore-sized membrane (Millipore, Billerica, MA, USA). The medium containing 20% FBS was added to the bottom well chambers. The cells were incubated for 48 h, then fixed with 100% methanol for 10 min and stained with 0.5% crystal violet for 20 min. Five fields of view per chamber were randomly selected for cell counting.

### 2.7. Western Blot

The target cells were lysed on ice using radioimmunoprecipitation assay (RIPA) lysis buffer with phenylmethylsulfonyl fluoride (PMSF) and a phosphatase inhibitor cocktail (100:1:1). The protein concentration was determined using a protein assay kit (Solarbo, Beijing, China). Sodium dodecyl sulfate polyacrylamide gel electrophoresis (SDS-PAGE) was performed on samples containing 30 µg of protein and then transferred onto a polyvinylidene difluoride (PVDF) membrane. The membranes were blocked with TBST containing 5% skim milk for 1 h and then incubated overnight with primary antibodies against FSH-R (1:500, ab75200, Abcam), E-cadherin (1:10,000, ab40772, Abcam), N-cadherin (1:1000, ab245117, Abcam), B-cell lymphoma-2 (BCL-2) (1:1000, ab32124, Abcam), BCL-2-Associated X (Bax) (1:1000, ab32503, Abcam), AMPK (1:5000, ab32047, Abcam), Phospho-AMPK (pAMPK) (Thr172) (1:2000, ab133448, Abcam), and GAPDH (1:5000, 10494-1-AP, Proteintech, Wuhan, China) at 4 °C. The membranes were then incubated with horseradish peroxidase-conjugated secondary antibodies (1:5000, Proteintech) for 1 h at room temperature. The protein bands were visualized by enhanced chemiluminescence (ECL, St. Louis, MI, USA). The images were obtained using an Amersham Imager 600 imaging system (General Electric Company, Boston, MA, USA), and GAPDH was the control. The ImageJ software program was used for the results analysis (NIH, Bethesda, MD, USA).

### 2.8. Immunofluorescence

The cell slides were fixed with 4% paraformaldehyde for 30 min after being washed three times with PBS for 5 min each. After blocking the cells with goat serum for 60 min at room temperature, the cells were then incubated with the anti-FSHR (1:100, ab113421, Abcam) primary antibody for an additional night at 4 °C. On the following day, the fluorescent secondary antibody was incubated at 25 °C for 1 h and washed three times with PBS. Then, it was stained with DAPI for 10 min and sealed with an anti-fluorescence quenching agent. A fluorescence microscope was used to observe the cells (OLYMPUS, Tokyo, Japan).

### 2.9. Animal Study

Four-week-old BALB/c nude mice were obtained from SPF (BEIJING) Biotechnology., LTD. The mice were acclimated and fed in a light cycle environment (25 ± 0.5 °C, 50–60% humidity). After acclimatization for one week, ISK cells were injected into the left axilla (approximately 1 × 10^7^/0.2 mL) of the mice. Three groups of mice were randomly assigned: (1) sham surgery (sham); (2) bilateral ovariectomy (OVX) with dietary oestradiol (E2) supplementation (OE); and (3) OVX plus E2 with FSH (30 IU/(kg, d), Merck, Kenilworth, NJ, USA) (OE plus FSH). Group (1) received a vehicle injection and was used as the control. Group (2) was given desiccated E2 powder (2.6 ppm) as a hormone replacement diet supplement (Bayer, Leverkusen, Germany) to eliminate the impact of E2. Group (3) received FSH intravenously and daily for two weeks after E2 was added for five days; then, the mice were killed. The tumor volume was calculated every week for five weeks using the formula: $volume=\left(\pi \times length\times width^{2}\right)∕6$ [21]. The animal experiments were approved by the Ethics Committee of the Shandong Provincial Hospital.

### 2.10. Statistical Analysis

The software SPSS 24.0 was used to analyze the data. The Student’s *t*-test or one-way analysis of variance (ANOVA) were used to determine statistical significance. A *p* value of <0.05 was regarded as statistically significant.

## 3. Results and Discussion


*FSHR is expresse*
*d in human endometrial cancer.*


We assessed FSHR expression in EC tissue and adjacent tissue using a validated specific antibody. Figure 1A shows that FSHR expressed in the EC tissue was markedly higher compared with adjacent tissue. At the protein level, we used the Hela cell as a positive control [22], and we found that the ISK and KLE cell lines expressed FSHR while the HEC-1A cell line did not (Figure 1B). Immunofluorescence (IF) consistently demonstrated that FSHR was expressed in the ISK and KLE cell lines (Figure 1C). In conclusion, we proved that FSHR is expressed in human EC tissue cell lines.


*FSH promotes cell proliferation and the migration of endometrial cancer.*


To explore the function of FSH in cell proliferation, we performed a colony formation experiment (Figure 2A), and the findings revealed that the medium containing FSH significantly increased the number of colonies, suggesting that FSH can increase the proliferative capacity of EC. Then, we demonstrated the effects of FSH on cell migration through cell scratch experiments (Figure 2B) and Transwell experiments (Figure 2C). The results showed that after 24 h and 48 h, the ISK cells’ capacity to migrate was significantly increased due to the higher FSH concentration (*p* < 0.01 and *p* < 0.01). The Transwell experiment also supported this conclusion.

Next, we explored the mechanism of increased cell proliferation and migration and detected the cell adhesion glycoproteins, E-cadherin and N-cadherin, which regulate cell migration [23], as well as BCL-2 and BAX, which are involved in apoptosis [14]. The WB results showed that E-cadherin was reduced as FSH levels increased (*p* < 0.01) and N-cadherin and BCL-2 increased significantly when the FSH concentration was 100 IU/L (*p* < 0.05 and *p* < 0.05). In contrast, the BAX expression had no clear association with FSH. With a rise in FSH concentration, the ratio of BCL-2/BAX also increased (*p* < 0.05) (Figure 2D). This further proved that FSH inhibits tumor cell apoptosis and encourages the invasion and metastasis of ISK cells.


*FSH inhibits the activity of AMPK in endometrial cancer cell lines.*


Our previous study confirmed that FSH stimulates AMPK Ser485 phosphorylation and inhibits AMPK Thr172 phosphorylation, thus inhibiting AMPK activity [24]. To prove the effect of FSH on the AMPK pathway in EC, we incubated Ishikawa cells with different dose gradients of FSH. Using WB, as seen in Figure 3, we found that pAMPK (Thr172) expression gradually decreased with increasing FSH concentrations (*p* < 0.05) and there was no significant change in AMPK expression, indicating that FSH inhibits the activity of AMPK. This is consistent with previous research.


*FSH promotes neoplasia of endometrial cancer in nude mice.*


To investigate how FSH influences endometrial carcinogenesis *in vivo*, we generated an animal model of the perimenopausal stage. Because of the FSH-E2 feedback regulatory loop, the animal model was required to maintain normal E2 levels to exclude the effect of E2 on endometrial carcinogenesis. As a result, we designed an ovariectomized (OVX) mouse model and supplemented it with E2 (OE). Additionally, in this condition, the mice were given exogenous FSH [25]. As shown in Figure 4A,B, we successfully constructed a xenograft model of human EC in BALB/c nude mice. Notably, there were no significant differences in the tumor weights and volumes between the sham group and the OE group. The tumor weight and volume in the OE plus FSH (0.90 ± 0.04 g; 864.99 ± 44.54 mm^3^) group prominently increased compared with those of the sham group (0.72 ± 0.08 g, *p* < 0.01; 693.17 ± 77.48 mm^3^, *p* < 0.05) and the OE group (0.72 ± 0.04 g, *p* < 0.01; 708.55 ± 25.24 mm^3^, *p* < 0.01). These results suggested that FSH can promote the proliferation of EC *in vivo*.


*Discussion*


We explored the role and related mechanisms of FSH in endometrial cancer (EC). Our results demonstrated that FSH acts in EC via FSHR, promoting the proliferation and migration of EC both *in vitro* and *in vivo*.

FSH functions by binding to its receptor, FSHR. FSHR is expressed in granulosa cells, placenta, and endometrium in females, and in testicular cells and the prostate in males [26]. Recent studies have discovered that extragonadal tissues express functional FSHR, such as adipose tissue, vessel endothelial cells [27], and bone tissue [26]. Our research confirms for the first time that FSHR is expressed in human EC tissues, and it indicated significant differences in FSHR expression between the tumor tissues and adjacent tissues from the same patient. At the cellular level, we demonstrated that FSHR is expressed in the ISK and KLE cell lines but not in the HEC-1A cell line. Many studies have reported differences in the expression of many receptors among these three cell types, but no reports have identified the specific properties of HEC-1A that are independent of the other two cell lines to explain the differences in receptor expression [28]. Therefore, we hypothesized that the reason why HEC-1A does not express FSHR may be attributed to their different cytological characteristics, which deserves further exploration.

Given the relationship between estrogen and EC, we excluded the role of estrogen to verify the effect of FSH on EC. EC can be classified into two types: estrogen-dependent and non-estrogen-dependent cancers [10]. Due to the high estrogen level of estrogen-dependent EC, we proposed a nude mouse model of perimenopause in our animal experiments by removing the effect of estrogen. Exogenous E2 was given to surgical OVX mice to keep E2 levels comparable to the sham mice. Due to E2 feedback in the pituitary gland, endogenous FSH levels were also stable under these conditions. Thus, exogenous FSH was used to control FSH levels without affecting E2 levels [25]. We killed the mice after five weeks and measured the tumor volumes and weights. The results confirmed that the tumor growth rates, as well as tumor volumes and weights, were significantly higher in the OE plus FSH group than in those of the other two groups, implying that FSH can promote the proliferation of EC in mice.

The role of FSH in the occurrence and progression of tumors has sparked widespread concern. Nicolae Ghinea concluded that FSH influences tumor angiogenesis and vascular remodeling by promoting tumorigenesis, invasion, and metastasis via FSHR on breast and kidney tumor vessels [29]. Song et al. verified that FSH can improve ovarian cancer cell proliferation by activating sphingosine kinase [30]. A previous study demonstrated that FSHR is expressed in the endometrium, and the expression level is significantly higher in the proliferative phase [31], suggesting that FSH can be of vital importance in endometrial hyperplasia. BCL-2 and BAX are tumor apoptosis-related factors. BCL inhibits apoptosis caused by a variety of cytotoxins, whereas BAX promotes apoptosis. Chen et al. confirmed that FSH increased the expression of BCL-2 while inhibiting BAX [14], which was consistent with our results. Cadherins belong to the cell adhesion molecule family. E-cadherin interferes with molecular adhesion and intracellular signal transduction, and it inhibits tumor formation and metastasis. On the contrary, N-cadherin contributes to local invasions and the metastatic spread of cancer cells [32]. Our study is the first to demonstrate that FSH decreases E-cadherin expression while increasing N-cadherin expression at the protein level in EC, implying that FSH may inhibit the migration of EC.

Mammalian AMPK is a heterotrimeric complex containing a catalytic subunit α and two regulatory subunits, β and γ, with each subunit containing two or three different isoforms [33]. AMPK activation is mediated by phosphorylation at threonine 172 (Thr172) in its α subunit and suppressed by phosphorylated at serine 485 (Ser485) by Akt, which reduces Thr172 phosphorylation [34]. High levels of AMPK pathway activity are associated with prognoses in glioma, breast cancer, and sarcoma. High glucose levels have been shown to reduce E-cadherin expression while increasing BCL-2 expression through the AMPK/mTOR/S6 pathway, and then facilitating the progression of EC [35]. Metformin reduces neuronal apoptosis by inhibiting BAX expression via the AMPK/mTOR pathway [36]. In lung cancer, TWIST2 can reduce the expression of N-cadherin while increasing the expression of E-cadherin, and it can inhibit tumor progression through the AMPK/mTOR pathway [37]. Our present study demonstrated that FSH inhibits apoptosis and promotes the cell proliferation of EC cells by up-regulating N-cadherin and BCL-2 and down-regulating E-cadherin and BAX. Therefore, we hypothesized that FSH-induced EC tumor growth is associated with changes in AMPK activation and E-cadherin, N-cadherin, and BCL-2 expression, which regulates proliferation, apoptosis, adhesion, and the invasion of EC cells, thereby leading to the tumorigenesis of the endometrium. We intend to examine how FSH affects downstream molecules via AMPK as part of our ongoing study.

## 4. Conclusions

The accumulation of organic pollutants can lead to an increase in FSH levels in the human body [9], which may harm human health. Our results demonstrate that FSH can promote endometrial carcinogenesis independent of estrogen. The increased FSH can affect the proliferation, apoptosis, adhesion, and invasion of tumor cells, thus promoting the occurrence and development of EC. Additionally, organic pollutants can increase the levels of FSH and may be risk factors for EC. The above conclusions provide us with a fresh strategy for the treatment and management of EC in the future.

## Figures and Tables

**Figure 1 ijerph-19-15344-f001:**
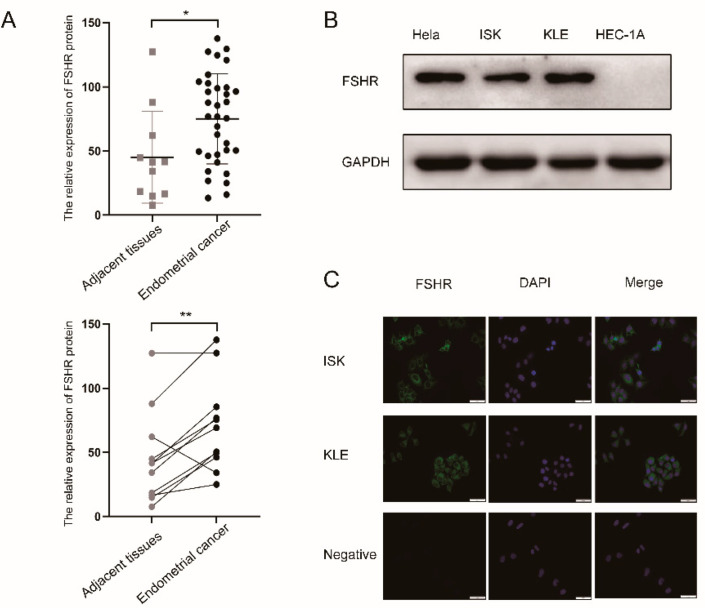
Expression of FSHR in EC tissue and cell lines. (**A**) Detection of the expression of FSHR in EC tissues and adjacent tissues by IHC. In the formula $Histochemistry\;scores=\sum \left(PI\times I\right)$, *PI* is the proportion of positive cells among all cells in the section and *I* corresponds to the color intensity of a section. (**B**) FSHR was detected by Western blot (WB) in the Hela cell line, as well as the ISK, KLE, and HEC-1A cell lines, and they were normalized with GAPDH. (**C**) The expression of FSHR in KLE and ISK was detected by IF. * *p* < 0.05, ** *p* < 0.01.

**Figure 2 ijerph-19-15344-f002:**
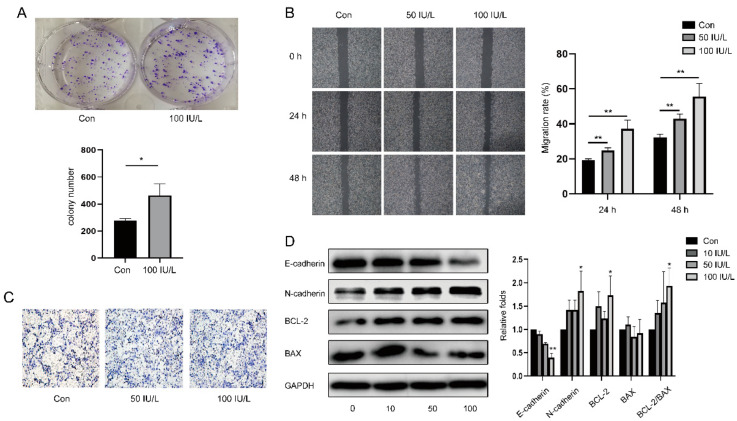
Effects of different doses of FSH on cell proliferation and migration in ISK. (**A**) Cells were treated with 100 IU/L FSH for 10 days and calculated. (**B**) Cells were treated with different doses of FSH (0 IU/L, 50 IU/L, and 100 IU/L) for 48 h, then cell migration rates were detected by cell scratch experiments at 24 h and 48 h using a (**C**) Transwell migration assay at 48 h. (**D**) E-cadherin, N-cadherin, BCL-2, and BAX protein levels were measured with WB, and GAPDH was the control. Relative band intensities were used to quantify E-cadherin, N-cadherin, BCL-2, and BAX protein expression levels. Data are expressed as means ± SDs. ** p* < 0.05, ** *p* < 0.01.

**Figure 3 ijerph-19-15344-f003:**
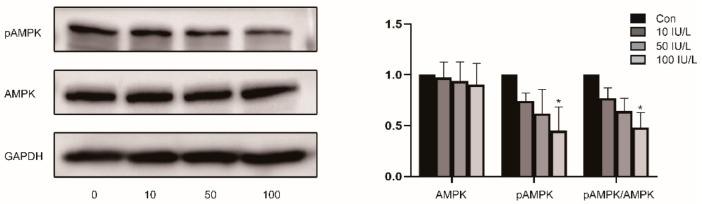
The effect of FSH on AMPK in Ishikawa cells. (Left) AMPK and pAMPK (Thr172) protein levels were detected by WB. (Right) Relative band intensities were used to quantify the level of AMPK and pAMPK (Thr172) protein expression. Data are expressed as means ± SDs. * *p* < 0.05.

**Figure 4 ijerph-19-15344-f004:**
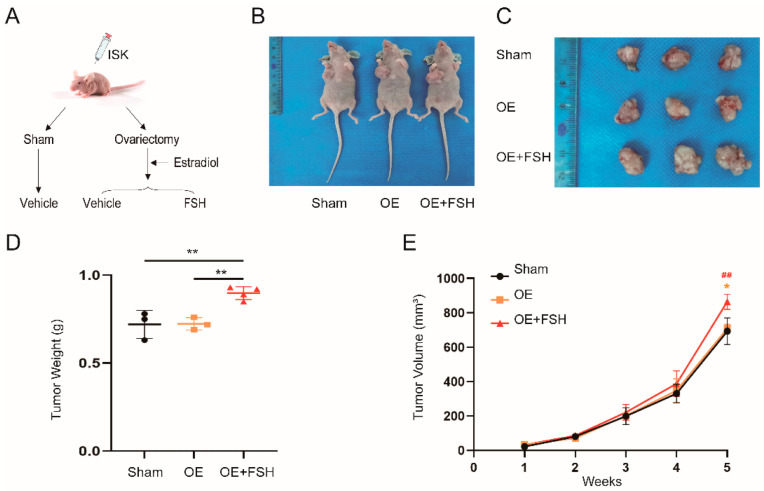
Effect of FSH on the neoplasia of endometrial cancer in nude mice. (**A**) Process diagrammatic drawing of BALB/c mice receiving different treatments. After acclimatization for one week, ISK cells were injected into the left axilla (approximately 1 × 10^7^/0.2 mL) of the mice. Three groups of mice were randomly assigned: (1) sham surgery (sham), (2) bilateral ovariectomy (OVX) with dietary estradiol (E2) supplementation (OE), and (3) OVX plus E2 with FSH (30 IU/(kg, d)) (OE plus FSH). Group (1) received a vehicle injection and was used as the control. Group (2) was given desiccated E2 powder (2.6 ppm) as a hormone replacement diet supplement to eliminate the impact of E2. Group (3) received FSH intravenously and daily for two weeks until (**B**) the mice were killed. (**C**) Tumors from the mice were collected and (**D**) weighed. (**E**) Weekly growth curves of the tumors and tumor volumes. The formula used was: $\mathrm{volume}=\left(\pi \times \mathrm{length}\times {\mathrm{width}}^{2}\right)∕6$. Data are expressed as means ± SDs. ** *p* < 0.01 and * *p* < 0.05 versus the sham group, and ^##^ *p* < 0.01 versus the OE group.

**Table 1 ijerph-19-15344-t001:** Endometrial cancer patient characteristics.

Number of Patients	34
Median age (years)	60
Age range (years)	35–80
**Pathology Diagnosis**	
Endometrioid adenocarcinoma	30
Endometrial clear cell carcinoma	1
Endometrioid adenocarcinoma with squamous metaplasia	3
**Grade**	
Grade 1	4
Grade 2	17
Grade 3	13
**FIGO Stage**	
IA	30
IB	2
II	2

## Data Availability

The datasets generated and/or analyzed during the current study are available from the corresponding author upon reasonable request.

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
