# Peer review of "Follicle-Stimulating Hormone Promotes the Development of Endometrial Cancer In Vitro and In Vivo"

_ijerph, 2022, doi:10.3390/ijerph192215344_

Round 1

Reviewer 1 Report

The research presented in this article confirmed that FSH promotes EC development in vivo. The research also established that FSH induces changes in the expression of E-cadherin, N-cadherin, and BCL-2 as well as inhibits AMPK activation in EC cell lines. Although the authors have generated informative results, several claims in this article were founded on insufficient data. My comments are as follows:

1.       My biggest issue with the manuscript is that its main conclusion “FSH promotes the proliferation and metastasis of EC through the FSHR-AMPK pathway” is not adequately supported by the evidence presented. First, in order to confidently make this claim, there should also be data showing that if we restore the function of AMPK pathway, FSH-induced tumor growth is diminished. Second, even if AMPK plays a role in EC, AMPK may not be the only mechanism by which FSH promotes EC development. The sentence “FSH promotes the proliferation and metastasis of EC through the FSHR-AMPK pathway” makes it sound like the AMPK pathway is the only pathway FSH uses to affect EC.

2.       In this article, the authors seem to suggest that the changes in E-cadherin, N-cadherin, and BAX in response to FSH levels are mediated through the AMPK pathway. Although it has been shown in the literature that E-cadherin, BCL-2, BAX, N-cadherin can be modulated through the AMPK/mTOR pathway, there is no direct evidence in this manuscript unequivocally linking the expression of E-cadherin, N-cadherin, and BAX with the activity of the AMPK pathway or showing that the expression of E-cadherin, etc. is downstream of AMPK. It is better to simply state that FSH induced tumor growth is associated with changes in AMPK activation and E-cadherin expression without claiming there is a causal relationship.

3.       One subtitle in the results section is “FSH inhibits the activity of AMPK in endometrial cancer.” I don’t believe this statement is well supported given it is only based on cell line data. I recommended changing it to “FSH inhibits the activity of AMPK in endometrial cancer cell lines.”

4.       There is some background on what endocrine disruptors and organic pollutants are as well as their sources in Discussion and Conclusion. Such background information should be moved to the beginning of the article and placed in the Introduction section to help the audience better understand the research in its context.

5.        I recommend the authors indicate that ISK, KLE, and HEC-1A are EC cell lines for clarity.

6.       At the end of the Introduction when authors were discussing the effect of metformin on AMPK and EC, I recommend the authors also discuss what prior research has found about the relationship between metformin and FSH.

7.       Please change the format of “FSHR is expressed in human endometrial cancer” in the beginning of the Results section for it to be consistent with the rest of the subtitles within Results.

8.       Please provide some rationale or hypothesis for why HEC-1A, unlike ISK and KLE, doesn’t show expression of FSHR.

9.       For the colony formation experiment in Fig. 2A, instead of only showing the picture, please also quantify the results through a graph.

10.   The authors used 10, 50, and 100 IU/L FSH in this research. For the results to be more meaningful, are these levels similar to physiologically level of FSH when mice and humans are exposed to organic pollutants?

11.   I recommend having an editor who is a native English speaker review the article.

Reviewer 2 Report

In this article, the authors reported how FSH affects the occurrence of EC. They found that FSH receptor was expressed both in EC tissues and cell lines by immunohistochemistry, immunofluorescence, and western blot. The manuscript is straightforward, well written, and concise and has clear results within the scope of a review article. Definitely deserves to be published and is a valuable contribution to the “International Journal of Environmental Research and Public Health”. Some minor comments need to be addressed before publication.

[1] “1. Introduction”, Lines 45-47:

“EC can be divided into estrogen-dependent and non-estrogen-dependent, with the former accounting for the majority, and its incidence is closely correlated to estrogen 6.”.

At that stage, the authors should report that from the therapeutic point of view, patients with advanced disease are treated with chemotherapy and less frequently with hormone therapy. Moreover, nowadays there is role for immunotherapy (pembrolizumab), monoclonal antibodies (trastuzumab) and VEGF inhibitors (lenvatinib). There are encouraging results showing efficacy and an acceptable safety profile in patients with advanced endometrial carcinoma. Lenvatinib plus pembrolizumab was granted accelerated approval for the treatment of patients with advanced endometrial carcinoma that is not MSI-H or dMMR, who have disease progression after prior systemic therapy, and who are not candidates for curative surgery or radiation.

Recommended reference: Makker V, et al. Lenvatinib Plus Pembrolizumab in Patients With Advanced Endometrial Cancer. J Clin Oncol. 2020;38(26):2981-2992.

[2] “1. Introduction”, Lines 59-62:

“Metformin is widely known as an AMPK inhibitor and has been used to treat metabolic diseases such as diabetes, obesity, and cancer. Particularly, AMPK is involved in the budding and development of EC 13 and suppresses cell proliferation and metastasis through the mammalian target of the rapamycin (mTOR) 14 .”.

Indeed, metformin may stimulate AMP-activated protein kinase activation with inhibition of the mTOR pathway. It is also known that programmed cell death is mediated by several protein factors that include the Bcl-2 protein family and the caspase group of cysteine proteases. The authors should mention that the upregulation of the Bax (Bcl-2 family member) increases the activity of the caspases and enhances the apoptotic activity. The inhibition of caspase-3 is included in the mechanism by which insulin promotes apoptosis. Apart from Bax, insulin downregulates Bad, which prevents programmed cell death. Many preclinical studies correlate Bcl-2-regulated apoptosis to metformin’s chemosensitizing effects. The chemosensitizing effect of metformin seems to be correlated with p53 function. In the presence of p53, metformin suppresses hexokinase II (glycolytic enzyme) and pyruvate dehydrogenase kinase (anti-apoptotic serine/threonine kinase).

Recommended reference: Ghose A, et al. Applications of Proteomics in Ovarian Cancer: Dawn of a New Era. Proteomes. 2022; 10, 16.

[3]Discussion”, Lines 247-248:

“High BMI and exposure to estrogen or synthetic estrogen compounds such as EDCs have been identified as risk factors for EC 8.”.

What about early menarche, late menopause, family history of endometrial cancer, polycystic ovarian syndrome and radiation exposure?

Recommended reference: Raglan O, et al. Risk factors for endometrial cancer: An umbrella review of the literature. Int J Cancer. 2019 Oct 1;145(7):1719-1730.

[4] Discussion”, Lines 290-293:

“Mammalian AMPK is a heterotrimeric complex, contains a catalytic subunit α and two regulatory subunits β and γ, with each subunit containing two or three different isoforms 28. AMPK activation is mediated by phosphorylation at threonine 172 (Thr172) on its α subunit and suppressed by phosphorylated at serine 485 (Ser485) by Akt, which reduces Thr172 phosphorylation 29.”.

There is also evidence that high levels of AMPK pathway activity are associated with better outcomes in glioma, breast cancer and sarcoma.

Recommended reference: Boussios S, et al. Ovarian carcinosarcoma: Current developments and future perspectives. Crit Rev Oncol Hematol. 2019;134:46-55.

Round 2

Reviewer 1 Report

The authors have generally addressed the referees' questions and concerns. However, in my comment #7, when I wrote "changing the format of “FSHR is expressed in human endometrial cancer” in the beginning of the Results section for it to be consistent with the rest of the subtitles within Results", I meant changing it to bold and italic, not changing the wording and sentence structure. What the authors have changed it to "FSHR expresses in human endometrial cancer" is not grammatically correct, so please change it back and modify the format (bold, italic, etc.)

Author Response

Dear Reviewer,

Thank you for your careful review of our manuscript. I'm so sorry that we didn't understand you correctly and made a simple error. We have changed the title back and modified the format to bold and italic. Once again, we would like to express our sincerest apologies.

We would like to express our great appreciation to you .

Yours sincerely,

Shuman Sheng